# Properties of Wood–Plastic Composites Manufactured from Two Different Wood Feedstocks: Wood Flour and Wood Pellets

**DOI:** 10.3390/polym13162769

**Published:** 2021-08-18

**Authors:** Geeta Pokhrel, Douglas J. Gardner, Yousoo Han

**Affiliations:** 1School of Forest Resources, University of Maine, 5755 Nutting Hall, Orono, ME 04469, USA; yousoo.han@maine.edu; 2Advanced Structures and Composites Center, University of Maine, 35 Flagstaff Road, Orono, ME 04469, USA

**Keywords:** wood flour, wood pellets, wood–plastic composites, transportation costs, physical properties, mechanical properties

## Abstract

Driven by the motive of minimizing the transportation costs of raw materials to manufacture wood–plastic composites (WPCs), Part I and the current Part II of this paper series explore the utilization of an alternative wood feedstock, i.e., pellets. Part I of this study reported on the characteristics of wood flour and wood pellets manufactured from secondary processing mill residues. Part II reports on the physical and mechanical properties of polypropylene (PP)-based WPCs made using the two different wood feedstocks, i.e., wood flour and wood pellets. WPCs were made from 40-mesh wood flour and wood pellets from four different wood species (white cedar, white pine, spruce-fir and red maple) in the presence and absence of the coupling agent maleic anhydride polypropylene (MAPP). With MAPP, the weight percentage of wood filler was 20%, PP 78%, MAPP 2% and without MAPP, formulation by weight percentage of wood filler was 20% and PP 80%. Fluorescent images showed wood particles’ distribution in the PP polymer matrix was similar for both wood flour and ground wood pellets. Dispersion of particles was higher with ground wood pellets in the PP matrix. On average, the density of composite products from wood pellets was higher, tensile strength, tensile modulus and impact strength were lower than the composites made from wood flour. Flexural properties of the control composites made with pellets were higher and with MAPP were lower than the composites made from wood flour. However, the overall mechanical property differences were low (0.5–10%) depending on the particular WPC formulations. Statistical analysis also showed there was no significant differences in the material property values of the composites made from wood flour and wood pellets. In some situations, WPC properties were better using wood pellets rather than using wood flour. We expect if the material properties of WPCs from wood flour versus wood pellets are similar and with a greater reduction in transportation costs for wood pellet feedstocks, this would be beneficial to WPC manufacturers and consumers.

## 1. Introduction

With global awareness in addressing environmental impacts and minimizing the emission of harmful pollutants, the wood composites industry is seeking more environmentally friendly materials for their products. With the utilization of recycled plastics and waste wood-based fillers, wood plastic composites (WPCs) manufacturing can be considered a green technology [1]. The concept of WPCs is not new where its modern application began in the 1970s and since the 1990s, the popularity of WPCs in North America has increased in decking and railing production [2]. WPCs are any composite products manufactured using plant (wood or non-wood) fibers, thermoplastic or thermoset resins and a small number of additives. WPCs offer the advantages of enhancing mechanical properties with higher strength and stiffness, decreases in density and abrasion compared to inorganic filler composites [3,4,5] and compared to solid wood, higher water and decay resistance, better acoustic performance, reduced weight, lower production costs and biodegradability [6,7]. They have wide applications in the automotive, and construction industrial sectors. 

In general, the processes followed in manufacturing WPCs are: extrusion, injection molding, and compression molding or thermoforming (pressing) [2]. Similarly, the newer production technologies are additive manufacturing processes based on extrusion processes and with laser sintering [2,8,9,10,11]. WPC compounding can be performed in an extruder (single-screw extruder, twin-screw extruder, conical twin-screw extruder or a combination type extruder) or high-speed mixers of the Henschel type. A Henschel mixer is a high intensity vertical mixer (causing the materials to move freely regardless of their size, density, friction coefficient, etc.) or melt type (used for making master batches with wax or materials with wax-like properties. Additive manufacturing or 3D printing consists of three general steps: usage of computer-aided design (CAD) to model the part, processing the model in 3D space through the slicing software, and with the G-codes printing and production of the part [12]. On an industrial scale, industries that manufacture WPCs either compound all the raw materials, i.e., wood filler, polymers, additives themselves or purchase pre-compounded WPC pellets. In 2019, the global market size of WPCs was USD $4.77 billion out of which North America (major application in decking) stood highest in market share with USD $2.12 billion [13]. Zion Market Research [14] has predicted that the WPCs global market in 2022 to reach to USD $8.76 billion in 2022. The expected mean annual growth rate is 12.3% between 2017 and 2022. 

Thermosetting plastics cannot be melted repeatedly once cured whereas thermoplastics can be repeatedly melted. Thermosetting resins include epoxy and phenolics whereas polyethylene (PE), polypropylene (PP), polyvinyl chloride (PVC) and polystyrene (PS) are the thermoplastic resins. PP has advantages of cost-effectiveness, lower material weight, better processability, resistance to extreme environmental conditions and reusability. Further, PP-based composites have applications in automotive, packaging and building materials [15]. Additives such as colorants, coupling agents, stabilizers, blowing agents, reinforcing agents, foaming agents and lubricants can be used based on the target area of application. In North America, commercial WPCs are mostly manufactured from wood shavings, sawdust or wood chips produced from industrial processes [5,16,17,18]. Different wood species can be used to manufacture WPCs. To effectively utilize the wood-based fillers in the thermoplastics, a basic knowledge in morphological and chemical characteristics is vital [19]. Depending on the woody material, different properties of the composite material can be obtained. However, from a commercial standpoint, usually materials that are conveniently available from the supply side are utilized rather than based on the characteristics of the fibers [20]. In Southeast Asia, hemp, ramie, kenaf and so on are mostly used as fillers as they are abundant there. However, in the US, species mostly utilized as fillers are the softwoods such as white pine, spruce, hemlock; and hardwood such as aspen. In general, the size of wood flour for manufacturing WPCs ranges from 1 mm–100 µm (18 mesh–140 mesh) [20]. However, the particle size of wood flour most preferable for composite products is in the range of 180–425 µm (40–80 mesh size) [21]. Larger filler sizes offer an advantage in terms of cost of pulverization and higher filler content in composite but with the disadvantage of lack of water resistance and difficulties in fabrication during injection molding. Similarly, smaller filler sizes offer the advantages in terms of mechanical properties and durability but disadvantages in terms of comminution costs and filler contents. The risk of a dust explosion increases for fine powders as well [20].

Equally important is another aspect focusing on the logistics and supply chain of the product, which is often a major factor deciding the final cost of the end product. Most of the time the manufacturers of WPCs purchase the raw material of wood flour from the wood mills that can manufacture wood flour or other authorized manufacturers of wood flour. As of now, there is no advancement in technology that can compress fluffy wood flour to ship over long distances. Wood flour because of its lower bulk density has a larger volume and standard truck trailers are not able to reach the maximum weight load limits (40 tons). The low density and higher volume of the wood flour make longer distance transportation disadvantageous in terms of cost. Being a fluffy material, its shipment costs over longer distances exceed the actual material price. The higher price in purchasing raw material consequently has a negative impact on the production, distribution, trade, and/or retail sale on manufacturing the WPCs. Motivated by this problem, the current study aims at exploring the utilization of a compacted wood flour, i.e., wood pellets to manufacture WPCs. Then, the physical and mechanical properties of the resulting composite products are studied. Wood pellets are easier to transport and handle as well. Production of wood pellets might have slightly more costs than wood flour. However, a comparative analysis of bulk density of wood flour (190–220 kg/m^3^) and wood pellets (700–750 kg/m^3^) reveals that roughly three and half times more wood pellets can be transported in a truck trailer than wood flour by weight. This suggests it is economical to transport wood pellets over longer distances. Very limited articles such as [22] have reported the application of wood pellets to manufacture WPCs. However, no research has focused on transportation of raw materials for WPCs manufacturing in a cost-effective way. The first part of this study presented the results on the characterization of the properties of wood flour and wood pellets produced from secondary processing mill residues [23]. Wood flour and wood pellets were made from four different wood species: Northern White Cedar (*Thuja occidentalis,* Eastern White Pine (*Pinus strobus*), Eastern Spruce-Balsam Fir (*Picea rubens-Abies balsamea)* and Red Maple (*Acer rubrum*). 

In this paper, discussion on WPCs manufactured utilizing wood filler as flour and pellets separately in a PP polymer matrix are presented. Utilizing wood pellets is a novel concept in the commercial manufacturing process of WPCs. Thus, a study of WPCs’ physical and mechanical properties through the application of wood flour and wood pellets separately as raw material source of wood feedstock are highlighted in this study.

## 2. Materials and Methods

### 2.1. Materials 

For the manufacturing of WPCs, the raw material based on wood, i.e., wood flour and wood pellets were prepared in the laboratory using local mill processing residues in Maine, USA. Part I of this paper series [23] has detailed information on equipment used and processing parameters followed during manufacturing of wood flour and wood pellets. The mill residues were clean (free from barks, adhesives, metals, etc.) and low in moisture content. They were grinded in a hammermill (Bliss Industries LLC, Ponca City, OK, USA) using a screen size of 0.5 mm. The produced wood flour was classified into different mesh sizes. Wood pellets made from 40-mesh wood flour was utilized as a wood feedstock in the manufacturing of WPCs. A pellet mill (Lawson Mills Biomass Solutions Ltd., North Wiltshire, PE, Canada) with a quarter inch thickness die was used for the production of wood pellets. Wood flour ground in hammermill had a relatively low moisture content. The low moisture of wood flour caused hindrance in the binding of particles to form pellets. Thus, water was added and mixed manually with the wood flour and the moisture content was maintained between 10–15% depending on the wood species. Under high temperature and pressure, pellets were formed from the wood flour. It should be noted that for each of the four wood species, the 40-mesh sized wood flour and the wood pellets pelletized from the 40 mesh-sized wood flour were utilized separately in the composite product manufacturing. Figure 1 below shows the 40-mesh wood flour and wood pellets made with 40-mesh flour for each of the wood species. Each scale bar is 2 cm in length. Similarly, Polypropylene (PP) was purchased from ExxonMobil Chemical Company ((Houston, TX, USA) with the melt flow index (MFI) of 20 g/min at 230 °C and a density of 0.900 g/cm^3^. This nucleated, medium MFI rate homopolymer is suitable for general injection molding purposes. The coupling agent maleic anhydride polypropylene (MAPP) was purchased from the SI Group Inc. (Schenectady, NY, USA) with a MFI rate of 115 g/10 min at 190 °C and density of 0.6 g/cm^3^. The application of MAPP offers advantage as a compatibilizer and adhesion promoter and thus improving the mechanical properties [24].

### 2.2. Manufacturing Process of WPCs

PP-based WPCs using wood fillers of wood flour and pellets separately were manufactured. For each of the wood species, 40-mesh wood flour and wood pellets pelletized from 40-mesh wood flour were utilized. As mentioned in the Introduction section, composites from 40–80 mesh wood fibers show better material properties. Similarly, the decking manufacturers mostly prefer using 40-mesh wood flour. Hence, wood feedstock of 40-mesh flour was utilized in this study. Initially, the wood flour and wood pellets of different wood species were dried in oven at 103 ± 2 °C for at least 24 h. The moisture content of the wood feedstocks were ensured to be below 1% during the manufacturing process. For each of the wood species, there were four different formulations with different percentages of wood filler, PP and MAPP additives by weight. A total of 16 different formulations were formulated to study the properties of WPCs. Table 1 below shows the four types of formulations for each of the wood species with the weight percentages of different raw materials:

Danyadi et al. [25] have reported larger wood content in the composite product leads to particle aggregation causing the lower strength of the WPCs so the wood filler of 20% was used. Similarly, Lu et al. [26] found the maximum values of tensile and flexural strength at 15% and 35% of wood particles by weight respectively. 

For producing the WPCs using wood pellets, the wood pellets before extruding were ground into powder using a knife grinder (C.W. Brabender Instruments, Inc., South Hackensack, NJ, USA) using a screen size of 7.96 mm. It should be noted that the wood pellets discussed in this study represents the ground wood pellets using a knife grinder. The 2nd and 4th formulations, as mentioned in Table 1, were applicable on utilizing the wood pellets as feedstock for each of the wood species. After grinding the wood pellets, each component was first mixed manually for the equal mixing of the raw materials. A Brabender Twin Screw Extruder (Messrs. C.W. Brabender Instruments Inc., South Hackensack, NJ, USA) with a diameter of 20 mm, screw length L/D 40, flight depth of 3.75 mm, and screw speed of 150 min^−1^ was used for the extrusion process. The temperature profile of the extruder was set at 200 °C for the heating zones. The frequency of feeder was 15 Hz. Each formulation was fed to the extruders’ feed throat. When the materials start to pass along the barrel, the plastic materials start to melt and results in compounding of the raw materials. The compounded materials are forced through a die to make strands of the composites. When the materials were falling from the feeder at a constant rate, the feed rate of the different formulations was also measured during extrusion. For this, the raw materials mixture from each formulation were collected in a small container and weighed after one minute. The WPC filaments after allowing them to cool were pelletized in the similar knife grinder (C.W. Brabender Instruments, Inc., South Hackensack, NJ, USA) used in grinding wood pellets using a screen size of 7.96 mm. Once the WPC pellets were ground, the pellets were then dried in oven at 103 ± 2 °C for at least 24 h before injection molding. Samples were injection molded using a Mini-Jector Injection Molder Model #55E (Miniature Plastic Molding, Solon, OH, USA) with a ram pressure of 17 MPa at 200 °C. They were then left in the molds for 10 s to cool. Molds used provided the samples with dimensions as specified in ASTM D638-14 (Type I) and ASTM D790-17 for testing of the properties. Before testing, the samples were conditioned for at least 40 h at 23 °C ± 2 °C and 50% ± 10% RH. For producing the WPCs using wood flour, the first step followed when using wood pellets filler, i.e., grinding of wood pellets using a knife grinder was not applicable. Besides this step, the other similar processes were followed. The 1st and 3rd formulations seen in Table 1 were applicable on utilizing the wood flour as wood feedstock for each of the wood species.

### 2.3. Testing of Physical and Mechanical Properties of WPCs

A Zeiss NVision 40 scanning electron microscope (SEM) (Carl Zeiss Microscopy, LLC, White Plains, NY, USA) and a capacity of up to 1.2 nm resolution was utilized to study the distribution of materials in the WPC samples. A Rotary Microtome (Warner-Lambert Technologies Inc., Buffalo, NY, USA) was used to make the flat and smooth cross-sections of the WPC samples. Coating of the samples was done with an Au/Pd conductive layer of 4 nm thickness before the SEM observations. The images were magnified 100×, with a surface area of 100 µm in a high vacuum 2.24 × 10^−6^ Torr, and the electron source voltage was 3 kV. The samples were also observed under a fluorescent microscope to observe the distribution more clearly. An Olympus BH2 fluorescent microscope (Olympus Scientific Solutions Americas Corp., Waltham, MA, USA) with a wide blue fluorescent filter: 450–480 nm excitation, mirror 500 nm, barrier filter 515 nm and a CHIU technical Corporation M-100 100W mercury illuminator was used. Samples for fluorescent microscopy were the thin wood polymer films prepared from the same rotary microtome used in making samples for the SEM. SHUR/Mount (Triangle Biomedical Sciences Inc., Durham, NC, USA) was used as a mounting medium and the samples were mounted properly on the coverslip. Thus, the WPC samples for observation in SEM were flat and smooth cross-sections and for fluorescent microscope were the thin films. Both the microscopes used a Zeiss Axiocam ERc 5 s camera and Zen Blue software (1.1.1.0, Carl Zeiss Microscopy, LLC, White Plains, NY, USA). Image J software (National Institutes of Health and the Laboratory for Optical and Computational Instrumentation, Madison, WI, USA) was run to study the wood fillers’ particle dispersion in the polymer matrix. The area of each particle was calculated using the Image J software. Since the particles had variation in area, the number of particles for each area range (on every 50 square microns) was differentiated.

The American Society of Testing and Materials (ASTM) Standard D792-20 Standard Test Methods for Density and Specific Gravity (relative density) of Plastics by Displacement (ASTM International, West Conshohocken, PA, USA) was followed for determining the density of each sample after injection-molding. For each sample, five specimens were considered for the density testing. The temperature of the water was maintained at 23 °C ± 2 °C. Any bubbles observed were removed. Density was derived from the specific gravity calculation. The formulas to calculate specific gravity and density are:(1)$$\mathrm{Specific}\;\mathrm{gravity}\;=\frac{a}{\left(a+w-b\right)}$$
where, *a* is the apparent mass of specimen, without wire or sinker, in air, *b* is the apparent mass of specimen (and of sinker, if used) completely immersed and of the partially immersed wire in liquid, and w is the apparent mass of totally immersed sinker (if used) and of partially immersed wire.
Density = Specific gravity · 997.5(2)
where, 997.5 is the density of water at 23 °C.

ASTM D 638-14 Standard Test Method for Tensile Properties of Plastics (ASTM International, West Conshohocken, PA, USA) was followed to determine the tensile properties of the WPCs. Tests were performed at room temperature of 23 ± 2 °C and 50 ± 10% RH. A universal testing machine Instron 5966 (Instron, Norwood, MA, USA) with a 10 kN load cell was used. A mounted extensometer in the Instron measured the elongation of the samples. The tensile test speed was set at 50 mm/min for breaking the specimen within 5 min. For each sample, 15 replicates were tested to report the average value.

ASTM D 790-17 Standard Test Methods for Flexural Properties of Unreinforced and Reinforced Plastics and Electrical Insulating Materials (ASTM International, West Conshohocken, PA, USA) was followed for measuring the flexural strength and modulus of elasticity. The room temperature of 23 ± 2 °C and 50 ± 10% RH was maintained for the testing. A universal testing machine Instron 5966 (Instron, Norwood, MA, USA) with a 10 kN load cell was used. The support span length was 52.8 mm with an average depth of the beam of 3.3 mm. The outer fiber strain rate was 0.01/min and the crosshead motion rate was 1.5 mm/min. For each sample, 15 replicates were tested and the average value was reported.

ASTM D 256-10 Standard Test Methods for Determining the Izod Pendulum Impact Resistance of Plastics (ASTM international, West Conshohocken, PA, USA) was followed to determine the impact strength of the samples. A Ceast Izod Pendulum Impact Tester (Ceast U.S.A., Inc., Charlotte, NC, USA) with hammer energy of 2.75 J was used for impact testing. Before testing, the samples were prepared following the recommended dimensions in the standard. The recommend length of the specimens was 63.5 ± 2 mm and depth of notch 10.16 ± 0.05 mm with the angle of the notch 45 ± 1°. Notches were prepared using a milling machine and the appropriate length was cut on a band saw. Fifteen replicates for each sample were tested to report the average value.

### 2.4. Statistical Analysis

A three-way Analysis of Variance (ANOVA) with a significance level of 0.05 was used for testing of the statistical significance in the means of the variables. The statistical association of type of wood filler, presence or absence of MAPP, and wood species with the density, tensile properties, flexural properties and impact properties of the WPC samples was analyzed. Here, type of wood filler, presence or absence of MAPP, and wood species are the three independent input variables and their two-way or three-way interaction on the output variables was studied. The dependent output variables are the physical and mechanical properties determined in the study.

## 3. Results and Discussions

For the WPCs manufactured using wood pellets and flour, the SEM images did not show significant differences in dispersion or distribution or uniformity of particles in the matrix of polymer. Lee et al. [27] mentioned that the quantitative determination of the fibers organization in the WPC profiles is difficult using SEM images. Thus, fluorescent images were also taken to study these parameters. Figure 2 below shows the fluorescent images of WPCs manufactured from four different species using wood flour and wood pellets in the presence of MAPP. Fluorescent images of WPCs for four different wood species with wood flour and wood pellets separately in controlled formulation are shown in Figure A1 of Appendix A. In the fluorescent images, the wood fillers can be clearly observed. The black portion represents the polymer matrix alone or the polymer matrix with MAPP. PP and MAPP could not be separated in the fluorescent images. These images showed that the particles are more uniform in size for wood flour utilized WPCs than the pellet samples. Similarly, the distribution of particles is similar for both WPCs made with flour and pellets but the dispersion was higher for those manufactured with pellets than flour. This could be attributable to the grinding action of the knife grinder to the pellets that created particles with less uniformity than the sieved 40-mesh wood flour. Analysis was done on the dimensions of wood pellet and wood flour particles in the polymer matrix. The major axis, i.e., length of the wood pellets particles ranged from finer particles of 4 microns to the larger particles of 427 microns and the minor axis, i.e., width ranged from 2 to 220 microns. The size range had a greater deviation. For the wood flour particles, the major axis ranged from finer sizes of 4 microns to larger 295 microns and the minor axis from 2 to 180 microns with lesser variation than the former case. Likewise, the average aspect ratio of the 40-mesh wood flour particles on the polymer matrix for cedar was 2.2, pine 2.1, spruce-fir 2.4 and of maple 2.3. For the ground wood pellets, the average aspect ratio of cedar was 2, pine was 2.2, spruce-fir 2.1 and the maple 2. The biggest wood particles appear more or less rectangle in shape whereas the smallest particles are more irregular in shape [28]. Because of high temperature and shear forces, wood fiber length is reduced but not the width during compounding and molding [29,30]. On comparing the hardwoods and softwoods, it can be clearly observed that the distribution and dispersion of particles is higher in softwoods than the hardwood maple. This might be because of the softwood fibers being flexible and hardwood fibers stiffer. Hardwood fibers are shorter (about 1 mm) and softwood fibers are longer than the hardwoods (about 3–8 mm). The discrepancies between hardwood and softwood fibers in terms of density, morphology and aspect ratio influences the dispersion of the fibers and consequently, the reinforcing for the polymer [31]. The color of WPC samples for each wood species was different correlating with the color of mill residues, wood flour and wood pellets. During the melt processing, the orientation and dispersion of fibers in the polymer is affected as well [30]. Mechanical behavior of the composites are greatly influenced by the consistency of the lignocellulosic materials concentration in the polymer [32,33]. Thus, on comparing the physical properties of wood pellets after pelletizing and then grounding operation with the original wood flour, the ground pellets had greater variation in sizes and were more bulky. The particle size variation of the ground pellets was smaller attributable to the usage of uniform size of the wood flour [34]. As the particles are smaller, the circularity of the particle sizes increases [35]. The same authors also suggested drier pellets produce finer particles. Studies have been conducted related to the particle shape of the ground pellets. Some authors have suggested the mill type influences [36] whereas others have suggested the material properties affect the particles’ shape [37]. Jet mills produce particles with the highest aspect ratio. Likewise, in a dry condition, the chemical constituents of wood such as cellulose, hemicellulose and lignin usually do not change until reaching a temperature of around 200 °C [38]. However, surface energy of the particles might be reduced due to the thermal friction encountered by the particles during the milling operation. Grinding can change the surface properties of particles [39]. 

In this study, pellets were produced from the wood flour without any mechanical treatments. Durable pellets can be produced from the steam explosion mechanism of the raw materials [40]. Thermal pretreatment of the raw materials by torrefaction produce pellets with lower in properties such as density, hardness and energy yield, but higher in hydrophobicity [41]. The production factors such as additives, die temperature, pressure and raw materials can affect the properties of wood pellets [42]. With these changes in properties of pellets changes the material properties of WPCs. Butylina et al. [22] observed the physical and mechanical properties of PP-wood fibers produced from commercially manufactured wood pellets in between the values of wood flour and heat-treated wood fibers. Further detailed research on understanding the properties of wood pellets under different treatments and then its impact on composites manufacturing through different production and grinding treatments are suggested.

The feed rate of the raw materials under different formulations is shown in Table 2. Either in control or MAPP formulations, the feed rate using pellets was approximately 1.5 times greater than the feed rate using wood flour mixed with PP alone or with MAPP. This effect is slightly higher for formulations with MAPP than the controls as the weight of MAPP used was slightly more than the PP used. MAPP also acts as a processing aid, i.e., like a lubricant thus contributing to an enhanced production rate. Pellets after being processed in a grinder were bulkier than the wood flour fibers. These results suggest that time and energy are reduced for the processing of WPC pellets in an extruder using wood pellets rather than wood flour. Nevertheless, this is a case during extrusion only. The additional steps of pelletizing the wood flour and grinding operations still needs to be considered. The ground pellets possessed increased fragmentation in the polymer after the grinding.

The two graphs in Figure 3 below show the dispersion of particles in WPCs based on area of the particles. From both the graphs, it can be observed for both wood flour and wood pellets feedstock WPCs, most of the particles falls below 50 square microns followed by 50–100 square microns. However, the WPCs made from pellets have a steeper curve than that of utilizing wood flour. This suggests that within the same range of particles area, a higher number of the pellet particles fall than the flour particles. This then, affects the dispersion of particles in the polymer. A similar trend was observed through the visualization of fluorescent images in Figure 2. From the images, it can be inferred that WPCs made from wood pellets had better particles’ dispersion than WPCs made from the wood flour. However, proper dispersion and distribution is always challenging for small particles in the polymer matrix [25,43]. The clustering of the particles usually can occur with particles having a smaller particle size and higher surface energy [44].

The average density of PP is 900 kg/m^3^. Figure 4 below shows the density of WPC samples of wood flour and pellets in different formulations. For all sample formulations, the density observed was higher than 900 kg/m^3^. For all cases, WPCs made from wood pellets with MAPP showed a higher density. The increase in density of WPCs made with wood pellets is because of ground wood pellets being denser than the wood flour used. Matuana and Stark [45] suggested that the compression of cell walls in the wood causes increase in density of the WPCs than the pure plastics. From our results, it can be indicated that the cell walls of pellets squeezed more than the flour. Similarly, in a microscopic level, the cell wall density of hardwoods and softwoods is almost similar that does not create significant differences in the density of composites. Results of three-way ANOVA showed that the *p*-value on the one-, or two- or three-way interaction of each variable (wood filler type, wood species, and additives condition) was greater than 0.05 in all conditions except the one-way interaction of wood filler. In the case of density, presence or absence of MAPP and wood species type or their interaction with the wood filler, does not make a notable difference. However, as explained before, filler type alone shows some significant differences as WPCs with pellets have a higher density than those with wood flour. On average, the density of WPCs for the wood flour controls was lower by 0.5% than the pellet formulations. Here, the density was 0.6% and 0.3% lower for the wood flour controls than the pellets for softwoods and hardwood, respectively. Likewise, in the wood flour–MAPP formulations, the average density was 0.6% lower than with pellet–MAPP formulations which was lower by 0.6% and 0.8% for softwoods and hardwoods respectively.

The interplay of wood feedstock and the polymer impacts the mechanical properties of WPCs. The processes used in manufacturing WPCs also impact the composite products’ mechanical behavior [46]. The physical and mechanical properties depend on the nature of wood filler such as the size and distribution of particles, the orientation of fibers, wood species and wood filler contents. Softwood pulps augment the tensile and flexural properties of WPCs as compared to the hardwood pulps [31]. However, Berger and Stark [47] reported PP composites from hardwoods performed better than the softwoods. Either with wood flour or pellets, the tensile and flexural properties of the samples containing MAPP were better than the PP controls. This is obvious as MAPP increases the adhesion between polymer and wood. Tensile flexural and impact properties are improved attributable to MAPP in composite of wood flour and PP [48].

Figure 5 conveys the results of tensile strength and modulus of the WPCs under different formulations. Similarly, values of the tensile strength at yield and break and % elongation at yield and break are tabulated in Table 3.

Results of the three-way ANOVA showed the three variables alone, i.e., wood filler, additive and wood species or their two-way or three-way interactions indicate some statistical differences in the tensile properties. There was minute statistical difference in the tensile values attributable to the two-way interaction of wood species and the MAPP additive. Tensile properties with or without MAPP did not vary much among the different wood species. Similarly, tensile modulus was not influenced much based on the wood filler type and the three-way effect of filler type, species and MAPP additive. The graph in Figure 5 shows, WPCs manufactured using pellets or flour had no significant difference in the tensile values of the composite products. Either for the MAPP or control formulations, the wood pellet samples show similar or significantly lower differences in the tensile properties. On average, the tensile strength of the wood flour controls was 0.6% higher than the pellet controls. Tensile strength of the wood flour controls was lower by 1.7% for the softwoods and higher by 7.7% for the hardwood compared to the pellet controls. Similarly, the average tensile strength of the wood flour–MAPP samples was 3.9% higher than pellet MAPP samples. The softwoods and hardwood showed increases of 3.4% and 5.5% respectively for the wood flour–MAPP samples compared to pellet–MAPP samples. The average Modulus of Elasticity (MOE) for the wood flour controls was 3.8% higher than the pellet controls. In this case, the MOE of the wood flour control formulation compared to pellet controls were lower by 0.4% for the softwoods and higher by 16.4% for the hardwood sample. The average MOE of the wood flour–MAPP was higher by 10.2% than the pellet–MAPP formulation. The MOE was higher by 8.5% and 15.4% for softwoods and hardwood respectively for the wood flour–MAPP compared to the pellet–MAPP samples. These results show that the tensile property values differ based on the wood species. Rogers and Simonsen [49] suggested the type of wood species can influence the parameters such as: roughness, feasibility to grinding, and porosity which might regulate the bonding with WPCs. Compared to the other species, the spruce-fir WPCs exhibited the maximum tensile properties for all formulations. This is attributable to the higher fragmentation of fibers in the PP matrix. Spruce-fir fibers possess the highest aspect ratio among the wood species examined. Higher aspect ratio is crucial in influencing the mechanical performance of the WPCs than the length of the particle [16,17,50,51,52,53]. Damage to the wood fiber is induced by the grinding process [54]. WPCs made of hardwood, i.e., maple pellets have the lowest MOE values among the wood species made from pellets. Neagu et al. [55] reported the correlation between lignin content and stiffness of the composite materials where they observed maximum stiffness for softwood kraft fibers. These results also show similarities in the findings as maple pellets have less lignin content compared to the other softwood pellets. 

Table 3 lists the tensile strength at yield and break and % elongation at yield and break for different formulations of the composites. The values show WPCs manufactured using pellets and wood flour did not have significant differences and any differences were small. Spruce-fir WPCs showed maximum strength compared to the other wood fillers on average. Likewise, the elasticity of the composites is indicated by the elongation at break. The percentage elongation at yield is in the range of 2–4% and a break in the range of 8–12% for the WPC samples. Between the pellets and wood flour WPCs, the values indicate minor differences. Compared to the maple and cedar, pine and spruce-fir have greater % elongation at yield and break. The tensile strength and modulus, and % elongation of the WPCs have a positive relation with the finer size of the particles [56,57].

In Figure 6, the flexural properties of WPCs made with wood flour and pellets in the presence or absence of MAPP are presented. From the three-way ANOVA, it was observed for the flexural properties the variables alone or their two-way or three-way interactions contribute to some significant differences in the bending properties except the one-way interaction of wood filler type. The wood filler either the wood flour or pellets did not influence the flexural strength. MOE was affected by the filler type by small statistical difference. MOE was not influenced by two-way effect of wood species and MAPP additive where, the MOE values with and without MAPP were similar among the different species. The average flexural strength for wood flour controls was lower by 1.3% than the pellet controls. Flexural strength of the wood flour controls was lower by 4% for softwoods and higher by 6.7% for the hardwood formulation compared to the pellet controls. Similarly, the average bending strength of wood flour–MAPP samples was 4.2% higher than the pellet MAPP samples. The softwoods and hardwood contributed to increase in bending strength by 4.1% and 4.35% respectively on flour–MAPP condition than the pellet MAPP samples. The average bending MOE for the flour–control case was 1.3% lower than the pellet-control case. In this scenario, the MOE for the flour–control formulation contrary to pellet-control formulation was lower by 7.5% for softwoods and higher by 17.5% for the hardwood. The average bending MOE for formulation with flour–MAPP was higher by 8.4% than the pellet–MAPP formulation. The MOE was higher by 8.7% and 7.6% for softwoods and hardwood respectively in flour–MAPP case than the pellet–MAPP case. Looking at the graphs of the bending tests in Figure 6, a similar trend to flexural properties existed for formulations with or without additives and with wood flour or pellets. Wood flour or wood pellets WPCs had similar or significantly fewer differences in flexural properties. In some cases, bending properties of WPCs made with pellets were higher than WPCs with wood flour. Wood species had a different impact to the bending properties than to the tensile properties. Compared to the rest of the wood species, red maple exhibited the maximum flexural strength using MAPP for both flour and pellets followed by spruce-fir. However, Pilarski and Matuana [58] mentioned flexural properties of WPCs produced from rigid PVC and HDPE matrices showed better performance for softwoods than the hardwoods. Higher values of tensile and flexural properties was shown by jack pine and black spruce than the white cedar for HDPE composite [59]. Similarly, the flexural modulus of elasticity of WPCs with red maple pellets was lower for both controls and MAPP formulations than the other species. This might be attributable to the correlation between lignin content and stiffness as concluded by Neagu et al. [55].

Typically, the presence of wood fibers in the body of WPCs causes crack initiation and subsequent failure. The graph in Figure 7 show the impact properties of the different WPC samples. Unlike to the tensile and flexural properties, the presence of MAPP did not contribute to a higher impact strength than in the control formulations. Compared to the tensile and flexural properties, the impact strength values for different wood species had a greater variance from the mean value. On average, WPCs of red maple showed better impact strength than the other wood species. The study conducted by Bledzki and Faruk [60], on the effect of wood filler geometry on the physico-mechanical properties of PP/wood composites, reported hardwood flour reinforced PP composites had a better impact strength than the other PP composites of different fillers (softwood fiber, long wood fiber and wood chips). However, higher impact strength of PP based composites for ponderosa pine was greater than the hardwoods: oak and maple [47]. The composite samples made with fine wood particles have poorer impact strength than with coarse-wood particles [61]. In our study, the softwood particles were finer than the hardwood. Because of this, on average WPCs from softwoods had a lower impact strength than the hardwood. Similar to the tensile and flexural values, there was not a difference in the impact values due to the filler type. The average impact strength for the wood flour controls was 8.3% higher than the pellet samples. At flour–controls, it was higher by 14.3% for softwoods and lower by 9.6% for hardwood than the pellet-controls formulation. For the wood flour–MAPP formulations, the impact strength value was 2.3% higher than the pellet MAPP formulations. For softwoods it was higher by 3.7% and for hardwood lower by 1.8% at flour–MAPP formulation than pellet–MAPP formulation. Results of the three-way ANOVA on flexural properties showed the variables alone or their two-way or three-way interactions contribute to some significant differences in the impact properties. The statistical analysis also revealed that the interaction of wood filler and MAPP additive did not influence the impact strength with a greater difference. It means in each WPC formulation with presence or absence of MAPP, the impact strength for each filler type did not have much variation.

Future research work on using wood pellets as a wood feedstock in different polymers, different formulations, changing the properties of wood pellets during manufacturing, etc. are recommended to understand more about the material for its efficient application.

## 4. Conclusions

The following conclusions are highlighted from the present study:The physical and mechanical properties of WPCs made from either the wood flour or wood pellets were similar.Distribution of feedstock was similar for both wood flour and pellet formulated WPCs. However, dispersion was greater for WPCs with pellets than with wood flour.MAPP improved the physical and mechanical properties of WPCs for each wood feedstock and wood species.On average, WPC samples of spruce-fir species possessed the best properties.

## Figures and Tables

**Figure 1 polymers-13-02769-f001:**
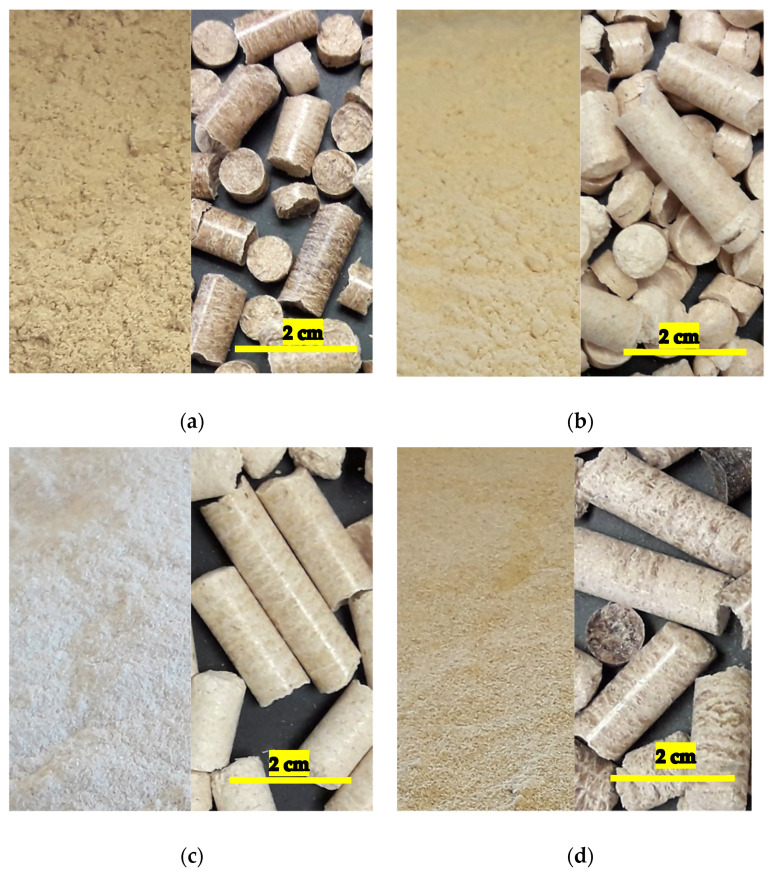
Wood flour and wood pellets used as wood feedstock in WPCs (**a**) Cedar, (**b**) Pine, (**c**) Spruce-fir, (**d**) Maple. (Image source for wood pellets: [23]).

**Figure 2 polymers-13-02769-f002:**
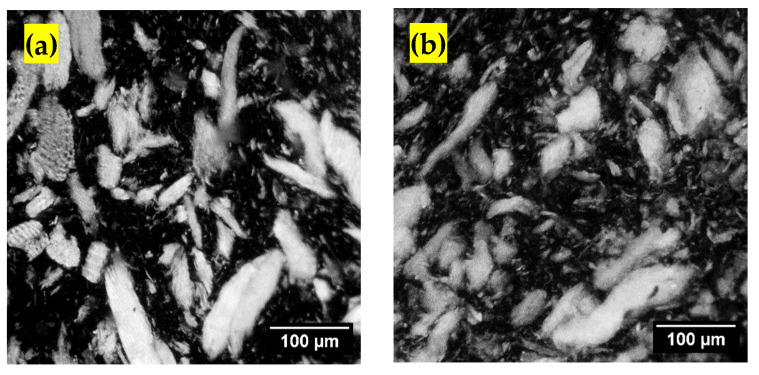
Fluorescent images of WPCs with MAPP (**a**) Cedar flour, (**b**) Cedar pellets, (**c**) Pine flour, (**d**) Pine pellets, (**e**) Spruce-fir flour, (**f**) Spruce-fir pellets, (**g**) Maple flour and (**h**) Maple pellets.

**Figure 3 polymers-13-02769-f003:**
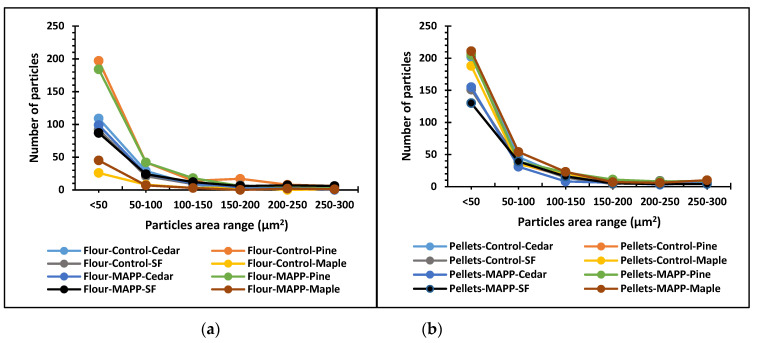
Area of wood particles in the polymer matrix (**a**) wood flour and (**b**) wood pellets.

**Figure 4 polymers-13-02769-f004:**
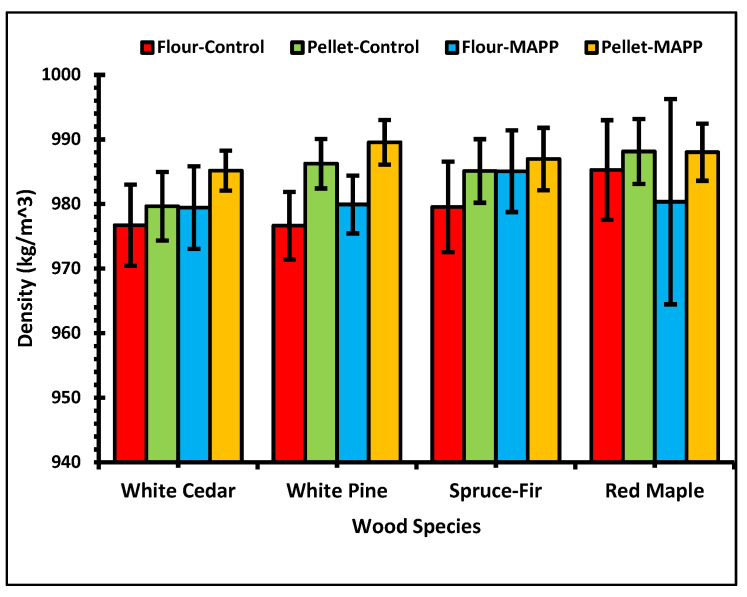
Density of WPCs.

**Figure 5 polymers-13-02769-f005:**
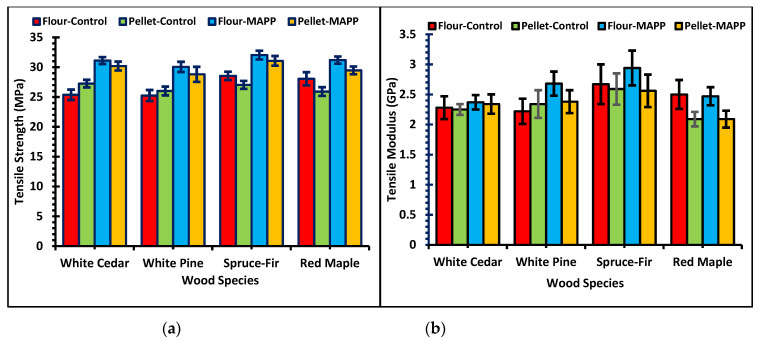
Graphs of (**a**) Tensile strength of WPCs and (**b**) Tensile modulus of WPCs.

**Figure 6 polymers-13-02769-f006:**
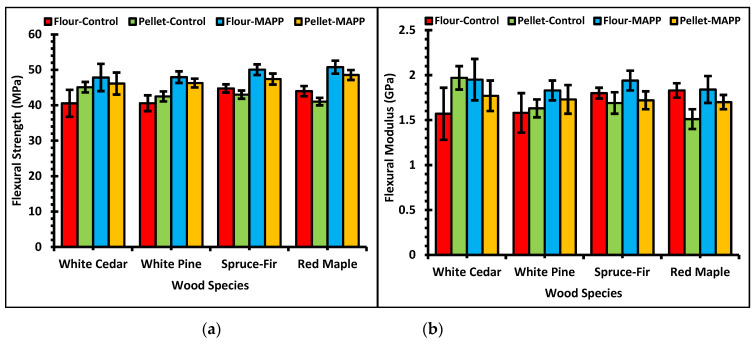
Graphs of (**a**) Flexural strength of WPCs and (**b**) Flexural modulus of WPCs.

**Figure 7 polymers-13-02769-f007:**
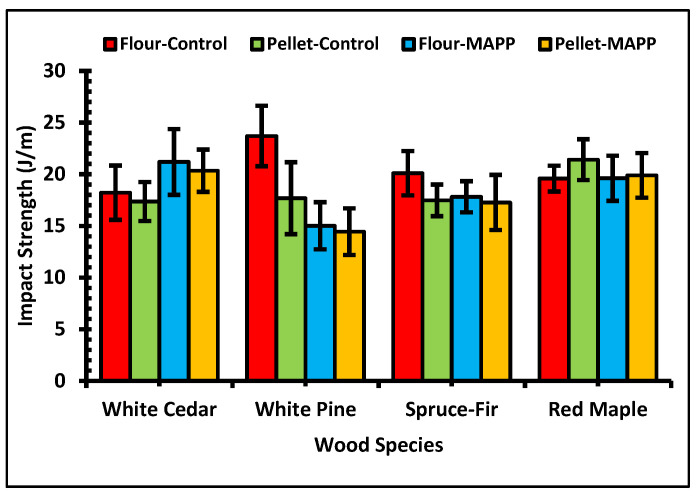
Impact strength of WPCs.

**Table 1 polymers-13-02769-t001:** Four different raw materials’ formulations of WPCs for each wood species.

Raw Materials	1st Formulation	2nd Formulation	3rd Formulation	4th Formulation
Wood flour	20%	0%	20%	0%
Ground wood pellets	0%	20%	0%	20%
PP	80%	80%	78%	78%
MAPP	0%	0%	2%	2%

**Table 2 polymers-13-02769-t002:** Feed rate (g/min) from the feeder of twin-screw extruder.

Wood Species	Controlled Condition		MAPP Condition	
	Flour + PP	Pellets + PP	Flour + PP + MAPP	Pellets + PP + MAPP
White Cedar	26.40	40.81	26.42	40.84
White Pine	22.03	39.62	22.05	39.65
Spruce-Fir	24.51	39.17	24.53	39.20
Red Maple	31.03	43.50	31.05	43.53

1 g of PP = 1.04 g of MAPP.

**Table 3 polymers-13-02769-t003:** Tensile strength and % elongation at yield and break for different formulations of WPCs.

Wood Species	Formulations	Tensile Strength at Yield (MPa)		Tensile Strength at Break (MPa)		% Elongation at Yield		% Elongation at Break	
		Mean	S.D.	Mean	S.D.	Mean	S.D.	Mean	S.D.
White Cedar	Flour–Control	14.35	0.82	14.82	0.49	2.77	0.40	10.62	1.10
	Pellets–Control	16.56	0.73	16.12	0.32	2.54	0.18	9.35	0.56
	Flour–MAPP	20.22	0.97	18.32	0.47	3.48	0.30	9.21	0.64
	Pellets–MAPP	17.81	0.65	17.60	0.50	3.13	0.26	9.31	0.53
White Pine	Flour–Control	15.43	1.38	14.82	0.64	3.32	0.32	12.72	1.16
	Pellets–Control	16.23	0.88	15.14	0.48	3.07	0.29	11.27	0.82
	Flour–MAPP	18.96	1.19	17.60	0.61	3.16	0.21	9.66	0.66
	Pellets–MAPP	18.69	0.76	16.89	0.74	3.33	0.47	9.43	0.75
Spruce-Fir	Flour–Control	17.74	1.02	16.72	0.52	3.28	0.52	9.73	0.90
	Pellets–Control	15.95	0.85	15.85	0.42	3.03	0.30	11.06	0.73
	Flour–MAPP	19.35	1.48	18.84	0.56	3.25	0.28	9.37	0.69
	Pellets–MAPP	18.77	1.74	18.34	0.48	4.05	1.09	10.50	1.10
Red Maple	Flour–Control	16.48	1.12	16.47	0.64	2.76	0.31	9.16	0.53
	Pellets–Control	15.14	0.82	15.27	0.47	3.11	0.59	10.28	0.90
	Flour–MAPP	18.64	0.53	18.20	0.44	3.42	0.83	10.16	1.01
	Pellets–MAPP	17.37	1.32	17.18	0.39	2.49	0.61	8.69	1.01

## Data Availability

All the necessary data is included within the article. Upon request, the data in the article can be available from the corresponding authors.

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
