# Peer review of "Properties of Wood–Plastic Composites Manufactured from Two Different Wood Feedstocks: Wood Flour and Wood Pellets"

_polymers, 2021, doi:10.3390/polym13162769_

Round 1
Reviewer 1 Report
The reviewed work entitled: Properties of wood-plastic composites manufactured from two different wood feedstocks: wood flour and wood pellets by Geeta Pokhrel, Douglas J. Gardner and Yousoo Han discuss the results of the work and PP-based WPC composites using various types of wood flour directly or indirectly obtained from pellets. The authors used 4 types of wood flour. They also produced a compressed form of wood flour in the form of pellets, which after grinding was used as a filler.
The reviewed paper is interesting due to the usage of a new form of filler obtained from the pellets of wood flour. However, I have a few questions about the manuscript:
There is no information on how pellets were prepared. Were there any additional means used for the production of pellets?
The methodology lacks information on what types of wood flour was used (this information is only in the description of fig1)
Why the produced WPC granulates weren’t dried before injection?
An analysis of the particle size of the original wood flour and that obtained after grinding the pellets should be performed to exclude the influence of processing conditions on the fragmentation of the filler in the polymer matrix (p. 7).
According to the authors, are the production costs of WPC from pellets lower than when using wood flour?
The authors claim: "These results suggest that time and energy are reduced for the processing of WPC pellets in an extruder using wood pellets rather than wood flour." (p.7.) Is it not the effect of increased fragmentation of the filler after grinding? How the samples were prepared for microscopic examination? Are they in the form of thin WPC polymer films or cryogenic sample fractures?
Author Response
"Please see the attachment."

Reviewer 2 Report
The background of the study is very well exposed, including WPC definition, manufacturing processes and advantages comparing to plastic composites with inorganic fillers or natural wood materials. The aspect of impact of filler’s particle size on the manufacturing process and WPC properties are highlighted, and even more prioritized, when logistics and supply chain of raw wood material are described, giving the possibility to the reader to think about disadvantages of wood flour as less appropriate for WPC if consider the transportation issues. The motivation of the study is very well explained and clear. Aim of the study was to investigate wood pellets as a compact form of wood flour for manufacturing of WPC. The lack of researches on transportation of raw material for WPCs manufacturing in a cost-effective way together with only one study on wood pellets for WPC has been given as a motivation and the novelty of the presented study.
2.1 Materials
Fig 1. Although size of the scale bar is given in the text referring to the figure, it must be included in the figure to make it self-describing.
The details of equipment used for manufacturing of pellets and process parameters should be given in Materials and Methods part.
2.2. Manufacturing process of WPCs
It seems unusual to find such discussion on methods chosen in Materials and Methods part, however, it gives better understanding and justification why such formulations were chosen for research.
What was the fraction size of ground wood pellets?
What physical or chemical impact on wood could be caused by pelleting and grounding operations in comparison to wood flour? If this is discussed in Part I of paper series, short explanation with reference would be great.
2.3 Testing of physical and mechanical properties of WPCs
Formula 1 and 2
The reviewer suggests to put symbols in formulas and explanatory text under the formula for each symbol instead of placing the whole explanatory text in the formula.
How fiber size of hardwood and softwood can be attributed to research, if the raw material for WPC was flour after 40-mesh? Do authors consider the effect of fiber properties on the properties of flour after being included in WPC? Please discuss, if and how wood particles can keep any fiber characteristic when are being very small in size and represent just a non-representative fragment of fiber? It would be great to consider chemical content differences of hardwood and softwood more.
Fig.2 It is hard to see the scale bar size. Please make letters (numbers) more visible and clearer.
Table 2. To what exactly does the note under the table refer to? Furthermore, sentence “*1 g of PP = 1.04 g of MAPP” is bit confusing, because earlier in the text specific formulations were given and MAPP was only 2% from formulation and according to the authors “formulations with different percentages of wood filler, PP and MAPP additives by weight”.
It is known and also discussed in the manuscript that particle size is one of the important factors influencing properties of WPCs. If authors wanted to compare the effect of shredded wood pellets and wood flour, more attention must be paid to the method of cutting/shredding of pellets and also to costs of this operation compared to the effect reached. So, the question should be answered – if the costs of energy needed for shredding of pellets does not exceed savings from transportation? How to evaluate level of comminution in terms of WPCs properties obtained? In this research it seems that researchers were lucky and obtained particle size distribution of shredded pellets was good enough to achieve the same properties as for WPCs with flour. However, reviewer suggest to investigate the effect of different mechanical treatments of pellets on the properties of WPC, maybe there is some optimal conditions to reach better properties of WPCs.
It would be great to calculate energy consumption needed for shredding the pellets, evaluate regarding possible transportation costs etc. to justify the advantage of this method (pellets vs flour).
In general interesting topic, although reviewer is in doubt, if this fits in journal Polymers. Journals with more focus to Materials, Economics, more industry related journals would be more appropriate.
Author Response
"Please see the attachment."

Round 2
Reviewer 2 Report
Dear authors, thank you very much for effort made to answer questions and expand discussion on presented study and topic in general. Manuscript has been improved and I suggest it to be published.